# FusionCell: Cross-Attentive Fusion of Layout Geometry and Netlist Topology for Standard-Cell Performance Prediction

**Haoyi Zhang**[1]  **Kairong Guo**[1]  **Bojie Zhang**[2]  **Yibo Lin**[1]  **Runsheng Wang**[1]

## Abstract

Standard cells form the building blocks of digital circuits, so their delay and power critically influence chip-level performance; yet characterization still relies on slow simulation sweeps, and many fast predictors ignore layout geometry, missing coupling and layout-dependent effects. The challenge is to jointly represent layout geometry and netlist topology so models capture fine-grained spatial details together with structural connectivity for accurate performance prediction. We introduce **FusionCell**, a dual-modality predictor that treats routed layout geometry and netlist topology as inputs and fuses them explicitly in a unified model. A DeiT encoder processes three-layer routed layouts, while a graph transformer models heterogeneous device/net graphs. The modalities are integrated through a **topology-guided** mechanism, where the netlist acts as a structural "map" to actively query relevant physical regions in the layout for joint geometric and topological reasoning. We build a 7nm dataset based on the ASAP7 PDK with over 19.5k cells spanning 149 types using automatic tools, targeting six metrics: signal rise/fall delay, transition, and power. Experimental results demonstrate that **FusionCell** reduces regression error (average MAPE 0.92%) and improves Spearman/Kendall ranking over baselines, while accelerating the characterization process by orders of magnitude compared to circuit simulation.

## 1. Introduction

Standard cells are the foundation of digital VLSI design (Weste & Harris, 2011; Kang & Leblebici, 2003), and

---
[1]School of Integrated Circuits, Peking University, Beijing, China [2]PicoHeart, Beijing, China. Correspondence to: Yibo Lin <yibolin@pku.edu.cn>.

*Proceedings of the 43$^{rd}$ International Conference on Machine Learning*, Seoul, South Korea. PMLR 306, 2026. Copyright 2026 by the author(s).

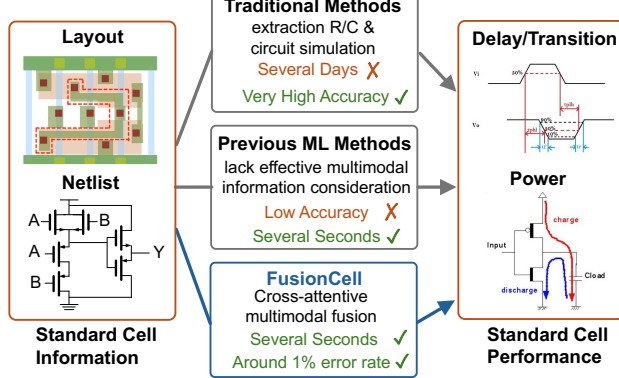

Figure 1. Comparison between traditional characterization, prior ML approaches, and FusionCell. Traditional flow (R/C extraction+simulation) is accurate but slow. Previous ML methods (e.g., vision-only or netlist-only) sacrifice accuracy for speed, often failing to capture layout effects or topological constraints. FusionCell achieves both high speed (milliseconds) and high correlation with golden tools via topology-guided fusion.

their performance (including delay and power) directly impact overall chip metrics. Such performance is jointly determined by layout geometry (which determines parasitic resistance and capacitance) and netlist topology (device connection, drive strength, IO fanout). As Figure 1 illustrates, standard-cell characterization traditionally involves a slow, multi-step pipeline: extracting resistance and capacitance (R/C) values from layout and running exhaustive simulation sweeps, which can take days for a full library. In contrast, FusionCell approximates this process in a single forward pass, delivering results in milliseconds—a $10^4 \times$ speedup—while maintaining high correlation with golden tools. At advanced nodes, Design Technology Co-Optimization (DTCO) pipelines spin out massive numbers of drive strength and layout variants to lift overall chip performance. Exhaustive characterization across these vast design spaces is no longer practical even with industrial automation (Klemme et al., 2020; Chen et al., 2025), making acceleration tools urgent.

A fast standard-cell evaluator is therefore needed that remains sensitive to layout-dependent R/C effects while still respecting netlist topology. Dropping layout underestimates layout-induced delay and power shifts; dropping netlist

misses topology constraints. The key is how to fuse layout geometry and netlist topology so the model can discriminate fine-grained layout differences and obey electrical connectivity, enabling rapid iteration of standard-cell layout quality to lift overall chip performance.

Some works (Ma et al., 2024; Liu et al., 2025; Cheng et al., 2024; Mallik et al., 2013) accelerate standard-cell evaluation via connectivity-centric graph neural networks (GNN (Yang et al.)) and graph transformers (Shehzad et al., 2024). They capture netlist structure yet omit explicit layout geometry, obscuring layout-induced R/C effects. Generative characterization libraries (Wu et al., 2024) and feature-driven regressors (Klemme & Amrouch, 2021) accelerate evaluation but lean on handcrafted descriptors and remain netlist-focused. ProtoCellLayout (Luo et al., 2025) introduces layout information but still models layout geometry in a specific graphs, lacking explicit alignment between netlist topology and layout geometry for fine-grained performance reasoning.

Beyond the standard cell itself, circuit representation work spans netlist-centric GNN/GTN surrogates and coarse layout-augmented graphs (Ma et al., 2024; Liu et al., 2025; Cheng et al., 2024; Luo et al., 2025), vision-only layout regressors that ignore net identity and detailed routing geometry (Zhao et al.; Zhu et al.), and multimodal or RTL-to-layout distillation aimed at higher-level designs (Wang et al.; Pei et al.; Wu et al.). At the standard-cell level, layout and netlist have to move in lockstep: the netlist defines the performance envelope, and layout-induced R/C effects decide where you land within. Therefore, keeping layout geometry explicitly tied to netlist topology is essential to preserve true connectivity rather than being misled by visual texture.

With the emergence of transformers (Vaswani et al., 2017), models such as Vision Transformer (ViT) (Dosovitskiy et al., 2021) and Data-efficient Image Transformer (DeiT) (Touvron et al., 2021) now surpass Convolutional Neural Networks (CNNs) (He et al., 2015) in capturing long-range dependencies, enabling finer-grained correlation across distant layout regions and thus supporting more comprehensive circuit-performance assessments. Moreover, graph transformers (Yun et al., 2019; Shehzad et al., 2024) contribute explicit edge encodings for heterogeneous circuit graphs, which capture richer structural context for downstream predictions. Despite their strong capability, a major challenge remains: designing layout and netlist representations that fully leverage the models' capability to capture geometric detail and graph structure, and fusing these modalities for robust standard-cell performance prediction.

This paper proposes **FusionCell**, a dual-modality standard-cell performance predictor that treats routed layout geometry and netlist topology as mandatory inputs, and integrates them through an explicit multimodal fusion framework to approximate golden characterization in a single forward pass. Intuitively, this approach is akin to using a schematic (netlist) as a map to locate specific components and connections in a satellite image (layout), ensuring that pixel-level details are interpreted in their correct functional context. FusionCell is built on the principle that layout-induced R/C effects must be interpreted under correct electrical connectivity, rather than learned independently or via late fusion. To this end, we encode multi-layer routed layouts (comprising metal routing layers M0, M1, and M2) using a DeiT backbone that preserves performance-critical metal and via geometry, and model netlists with a graph transformer on heterogeneous device–net graphs that explicitly distinguishes connectivity and correlation. The two modalities are fused such that netlist topology guides the aggregation of geometric evidence, preventing modality collapse and vision-only shortcuts; this is implemented via **topology-guided** graph-query/image-key cross-attention. FusionCell targets six key delay and power metrics (*signal rise/fall delay, transition, and power*). We further generate a 7nm dataset based on the ASAP7 PDK with over 19.5k cells spanning 149 types using automatic standard-cell layout generation tools (Guo & Lin, 2025). On this benchmark, compared with ProtoCellLayout (Luo et al., 2025), FusionCell achieves lower error across all targets (average MAPE 0.92%) and consistently stronger cell-variant ranking correlation with respect to golden results, demonstrating the effectiveness of explicit layout–netlist fusion.

Our contributions are summarized as follows:

- We propose **FusionCell**, a dual-modality standard-cell performance predictor that jointly models routed layout geometry and netlist topology through topology-guided multimodal fusion, achieving accurate and ranking-stable performance prediction.

- We propose a geometry-preserving representation and encoding strategy for routed layouts that retains R/C-critical metal and via information across multiple routing layers, enabling fine-grained spatial reasoning from layout geometry.

- We introduce a topology-aware netlist modeling strategy based on heterogeneous device–net graphs, which explicitly captures both device–net connectivity and net–net correlation, enabling electrically meaningful structural representation for standard-cell performance prediction.

- We generate a 7nm dataset based on the ASAP7 PDK ($> 19.5$k cells spanning 149 cell types) using an automatic standard-cell layout generation tool (Guo & Lin, 2025). On this benchmark, FusionCell achieves an average MAPE of 0.92% across six key delay and power

targets and improves cell-variant ranking correlation over vision-only and prior multimodal baselines.

## 2. Preliminaries

### 2.1. Standard cell characterization

The standard-cell characterization takes the layout and the netlist as inputs and determines delay and power under various PVT and load/slew conditions. The flow first extracts R/C values from the layout via a field solver that numerically solves the underlying partial differential equations (PDEs). Then the R/C values will be back-annotated into the netlist, and finally runs circuit simulation (which is an ordinary differential equation (ODE) solver) to populate the final standard cell metrics (e.g. delay, transition, power). Repeating this across thousands of layout variants slows the overall chip design flow (Klemme et al., 2020; Chen et al., 2025). A fast yet accurate standard cell performance predictor would directly improve design efficiency and overall performance, which typically must consider both layout and netlist together.

### 2.2. ML-driven standard cell characterization

Handcrafted descriptors and statistical fits (Klemme & Amrouch, 2021; Klemme et al., 2020; Mallik et al., 2013; Wu et al., 2024) accelerate characterization but stay netlist-focused, missing layout-induced R/C effects and corner scaling. Graph surrogates improve connectivity modeling: netlist GNNs (Ma et al., 2024; Dong et al.; Shi et al.) and GTNs (Liu et al., 2025; Cheng et al., 2024) capture stack depth and IO fanout, yet still neglect layout geometry. ProtoCellLayout (Luo et al., 2025) injects layout cues by projecting routes to graph edges, but the geometry remains coarse and loosely aligned to the netlist. Beyond single cells, circuit-level regressors expose similar gaps. Vision-only layout models for larger blocks (Zhao et al.; Zhu et al.) operate on rendered textures without explicit net identities or detailed routing geometry. Cross-stage distillation from RTL or schematic to layout (Wang et al.) and multimodal circuit representation learning (Pei et al.; Wu et al.) transfer functional priors across modalities, yet methods tuned for higher-level digital or analog circuits do not transfer cleanly to standard cells where dense metal/via effects dominate. Layout geometry and netlist topology must move together—layout drives R/C effects, netlist sets the performance envelope—so we keep netlists mandatory, layouts explicit, and fuse them via cross-attention to let connectivity guide spatial interpretation.

### 2.3. Problem Formulation

FusionCell aims to accurately predict standard-cell performance including both delay and power metrics based on routed layout geometry and netlist topology. Each cell provides: (1) a multi-layer routed layout tensor $L$ encoding metal/via geometry that drives R/C effects, (2) a heterogeneous netlist graph $G$ with device/net nodes encoding drive strength (Width/Length) and typed edges (device–net connectivity and net–net coupling), and (3) ground-truth performance labels $y \in \mathbb{R}^6$ (signal rise/fall delay, transition, and power). The regressor must use both modalities to avoid shortcuts (texture without connectivity or connectivity without R/C effects) and should be judged by absolute error (e.g., MAPE) and ranking quality (e.g., Spearman/Kendall) across cell variants.

**Problem 1** (Layout–Netlist Performance Prediction). Given a routed layout tensor $L$, a netlist graph $G$, and targets $y \in \mathbb{R}^6$, learn $f_\theta$ to produce $\hat{y} = f_\theta(L, G)$ with low regression error while preserving the rank order of cell variants; both $L$ and $G$ are mandatory inputs.

## 3. Algorithm

We present **FusionCell**, a fast yet accurate standard-cell performance predictor that jointly reasons over routed layout geometry and netlist topology by leveraging **topology-guided** cross-attentive multimodal fusion. Unlike prior approaches that treat layout and netlist independently or fuse them only at the representation level (Luo et al., 2025), FusionCell is explicitly designed to ensure that *geometric evidence is interpreted under correct electrical connectivity*. Figure 2 provides an overview of the architecture and information flow.

### 3.1. Overview and Design Principles

The performance of a standard cell is jointly determined by two tightly coupled factors: (i) **layout geometry**, which governs wire resistance, capacitance, and signal coupling, and (ii) **netlist topology**, which defines electrical connectivity, device roles, and signal propagation constraints. Layout geometry alone is insufficient, as visually similar routing patterns may correspond to different electrical functions; netlist topology alone is also insufficient, as identical netlists may exhibit different performance due to layout-dependent R/C effects.

FusionCell is designed around three principles: (1) preserving R/C-relevant layout geometry, (2) respecting electrical structure in netlist representation, and (3) enforcing topology-guided multimodal fusion so that layout geometry is explicitly queried by electrical nodes, rather than naively concatenated or fused symmetrically. This design reflects the physical intuition that *netlist topology defines the performance envelope, while layout geometry determines the precise variation within that envelope due to layout effects*.

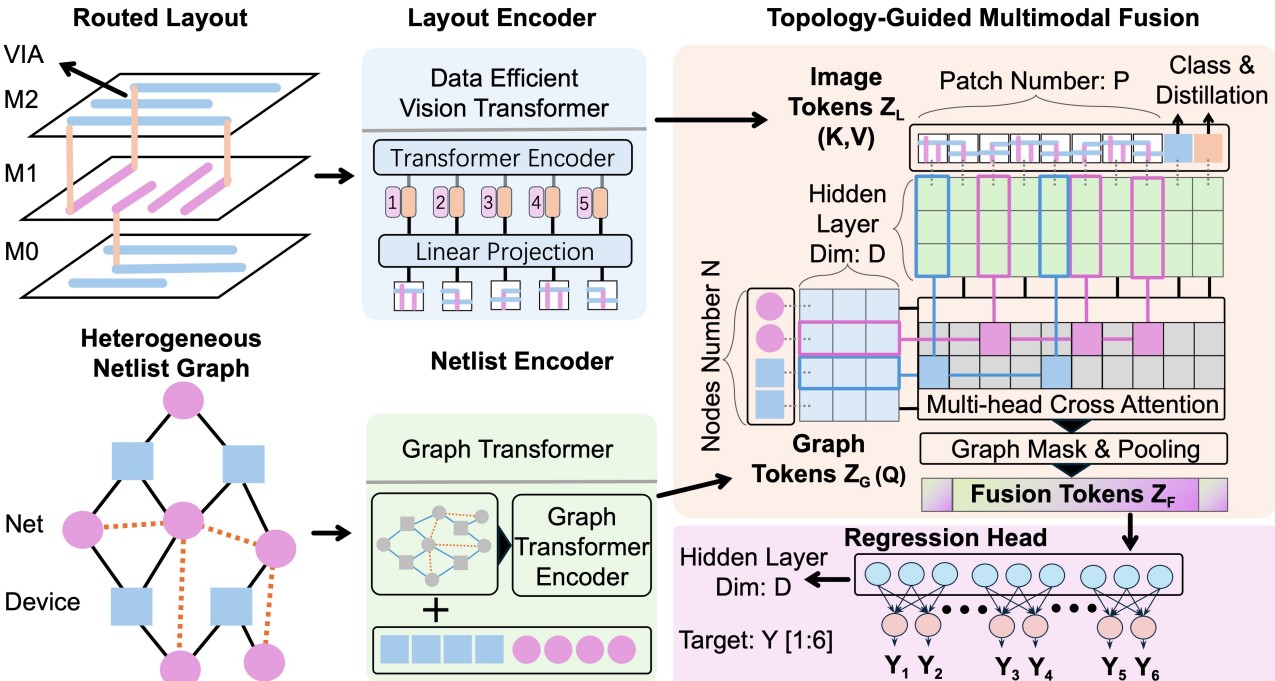

*Figure 2.* **Overview of FusionCell and topology-guided multimodal fusion.** FusionCell predicts standard-cell performance (including delay and power) from routed layout geometry and netlist topology. (Middle Up) A DeiT encoder processes three-layer routed layouts (M0/M1/M2) into layout tokens $Z_L$, including patch, class, and distillation tokens. (Middle Down) A graph transformer encodes the heterogeneous netlist—comprising device and net nodes with distinct edge types—into graph tokens $Z_G$, effectively capturing both local device–net connectivity and global net–net correlations. (Right) In the *Topology-guided multimodal fusion* stage, these structure-aware graph tokens act as queries to attend to the spatial layout tokens (keys/values), enabling the model to dynamically retrieve geometric details relevant to each electrical node. Finally, the fused tokens are aggregated via mask-aware pooling into a unified representation $Z_F$, which is passed to a Multilayer Perceptron (MLP) regression head to predict the six key delay and power metrics.

## 3.2. Layout Representation

The layout of a standard cell can be viewed as a multi-layer image where each layer corresponds to a specific metal plane used for signal routing. Physically, wires are geometric rectangles (polygons) placed on these layers to form electrical connections. The length, width, and spacing of these rectangles directly determine the parasitic resistance and capacitance (R/C) that govern circuit performance. To capture these effects, we treat the routed layout as a multi-channel image tensor, applying domain-specific preprocessing to preserve electrically significant details.

Routed layouts are rasterized into fixed-resolution tensors of size $H \times W$ with three channels corresponding to the primary routing layers (M0, M1, M2). In this visual representation, each channel encodes a 2D slice of the 3D metal stack: M0 is the lowest layer closest to devices, while M1 and M2 typically handle horizontal and vertical signal routing respectively. Pixels are populated with normalized Net IDs rather than simple binary occupancy; this ensures that connected geometries sharing the same electrical potential are represented by consistent semantic values across layers. To capture vertical connectivity (vias) without adding sparse channels, we apply an inter-layer projection strategy: 3D

vertical links are spatially expanded onto the adjacent 2D metal planes. This preserves the 3D connectivity structure within a compact 3-channel image format suitable for standard vision backbones. Since standard cells vary in aspect ratio, we align all cell layouts along their longer dimension and center them within the tensor canvas to minimize padding artifacts.

**DeiT Encoder.** The preprocessed layout tensor $L \in \mathbb{R}^{3 \times H \times W}$ is encoded using a Data-Efficient Vision Transformer (DeiT) (Touvron et al., 2021) backbone. The image is split into fixed-size patches (e.g., $16 \times 16$), which are linearly projected into patch embeddings. A learnable class token and a distillation token are prepended to the sequence. The tokens pass through a stack of transformer encoder layers with multi-head self-attention (MSA) and feed-forward networks (FFN). The output is a sequence of spatial layout tokens:

$$Z_L = \text{Enc}_{\text{layout}}(L) = [z_{\text{cls}}, z_{\text{dist}}, z_1, \ldots, z_P] \in \mathbb{R}^{(P+2) \times d},$$

where $P$ is the number of patches, and $d$ is the hidden vector dimension, representing the embedding size used to store information for each patch. Compared to CNNs, the transformer's global self-attention mechanism is particularly

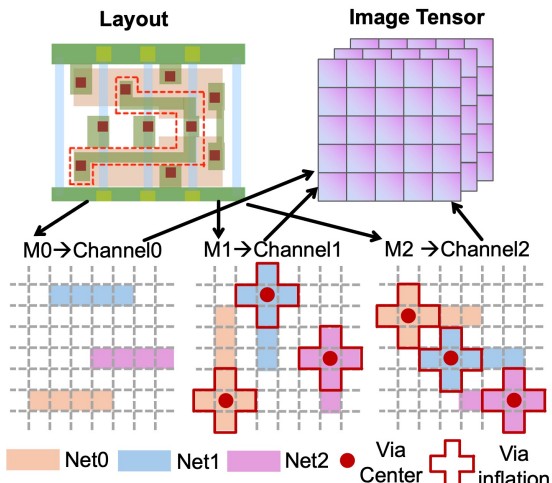

*Figure 3.* **Layout tensor encoding.** Routed layouts are rasterized into three-channel tensors (M0/M1/M2) with long-edge alignment. Preprocessing includes via spillover and neighborhood expansion to reveal coupling-relevant patterns.

well-suited for modeling long-range distributed effects, such as the IR drop across a long power rail or the cumulative coupling capacitance along a signal net that spans the entire cell width.

### 3.3. Netlist Representation

The netlist defines the logical function and the fundamental electrical constraints of the cell. We represent the netlist as a graph where the topology dictates the flow of information.

**Heterogeneous Graph Construction.** We parse the circuit netlist into a heterogeneous graph $G = (V, E)$ with two distinct node types: 1) **Device Nodes** ($V_D$) representing functional components (e.g., transistors), initialized with attributes such as type and drive strength; 2) **Net Nodes** ($V_N$) representing electrical connections, initialized with connectivity degrees and IO flags. Feature vectors are padded to align dimensions across types. The edges $E$ capture interactions: 1) **Connectivity Edges** ($E_{\text{conn}}$) represent physical wiring between devices and nets; 2) **Correlation Edges** ($E_{\text{corr}}$) connect structurally related nets, adding a higher-level view of signal flow and potential interference.

**Graph Transformer Encoder.** Standard GNNs (like GCN or GAT) typically perform local message passing, which limits their ability to capture long-range interactions in a few layers. In standard cells, signal propagation often involve spatially or topologically distant components. To effectively model these global dependencies and enable direct interaction between any pair of nodes (crucial for capturing holistic circuit behavior), we employ a graph transformer. Let $H^{(l)} = \{h_1^{(l)}, \ldots, h_N^{(l)}\}$ be the node features at layer $l$. The update rule for a node $u$ involves computing attention

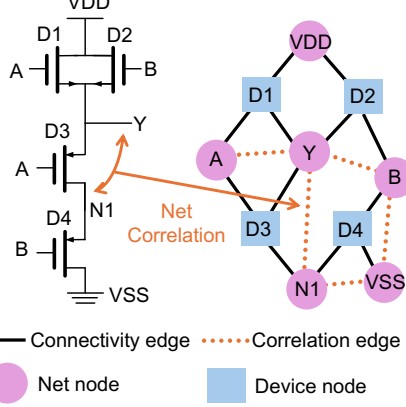

*Figure 4.* **Heterogeneous netlist construction.** Netlists are parsed into heterogeneous graphs with distinct device/net nodes. Edges are typed to distinguish physical connectivity (Device-Net) from structural correlation (Net-Net).

scores with all other nodes $v$:

$$\alpha_{u,v}^{(h)} = \text{softmax}\left(\frac{Q_u K_v^{\top}}{\sqrt{d}} + \phi(u, v)\right),$$

where $\phi(u, v)$ is a structural bias term. In our design, this bias is crucial for encoding the heterogeneity of the graph. We define:

$$\phi(u, v) = B_{u,v}^{\text{mask}},$$

where $B_{u,v}^{\text{mask}}$ enforces graph connectivity, allowing attention only between connected nodes (including both connectivity and correlation edges). This structural bias ensures that message passing respects the underlying circuit topology while maintaining the global receptive field characteristic of transformers. The final graph tokens $Z_G \in \mathbb{R}^{N \times d}$ are obtained after $N_{\text{layers}}$ layers of transformer encoding, containing rich structural information for every device and net in the cell.

### 3.4. Topology-Guided Multimodal Fusion

The core innovation of FusionCell is the *topology-guided* fusion strategy. As discussed, layout and netlist are not independent views; the netlist provides the functional "skeleton," and the layout provides the R/C "flesh." Symmetrical fusion strategies, such as naive concatenation, would lose the fine-grained correspondence between a specific net and its surrounding metal geometry.

**Graph-Query, Layout-Key Attention.** We propose to use the graph tokens $Z_G$ as the anchor to query the layout tokens $Z_L$. This is implemented via a multi-head cross-attention mechanism:

$$Q = Z_G W_Q, \quad K = Z_L W_K, \quad V = Z_L W_V.$$

The cross-attention output $\tilde{Z}_G$ is computed as:

$$\tilde{Z}_G = \text{softmax}\left(\frac{QK^{\top}}{\sqrt{d}}\right) V.$$

Intuitively, for each net node $u$ in the netlist (e.g., the output net "Y"), the attention mechanism scans the entire layout $Z_L$ to find patches that are visually relevant to "Y". Since the layout encoder preserves spatial information, the model can learn to attend to the specific metal tracks and vias that implement net "Y". This asymmetric design enforces a strong inductive bias: *we only care about layout features insofar as they affect the electrical components.* This prevents the model from overfitting to background texture or irrelevant silicon areas.

## 3.5. Prediction Head and Training Objective

The fused representation $z_{\text{fused}}$, obtained by pooling the updated graph tokens $\tilde{Z}_G$, is passed to a regression head, which is a simple Multilayer Perceptron (MLP) with GeLU activation and Layer Normalization:

$$\hat{y} = \text{MLP}(z_{\text{fused}}) \in \mathbb{R}^6.$$

The six output dimensions correspond to the standardized values of: 1) **Rise Delay:** Propagation delay for $0\rightarrow1$ output transition; 2) **Fall Delay:** Propagation delay for $1\rightarrow0$ output transition; 3) **Rise Transition:** Output signal slew rate (10%–90%) for rising edge; 4) **Fall Transition:** Output signal slew rate (90%–10%) for falling edge; 5) **Rise Power:** Dynamic energy consumed during rising transition; and 6) **Fall Power:** Dynamic energy consumed during falling transition.

**Loss Function.** We train the model end-to-end using Mean Squared Error (MSE) loss on the standardized targets:

$$\mathcal{L} = \frac{1}{B} \sum_{i=1}^{B} \|\hat{y}_i - y_i\|_2^2,$$

where $B$ is the batch size. We do not use auxiliary ranking losses (like Pairwise Ranking Loss) because our experiments show that a well-optimized regression objective naturally yields sufficient ranking correlation for cell selection tasks. The standardization of targets is crucial because power values (in fJ) and delay values (in ps) can differ by orders of magnitude. Using z-score normalization ensures that all tasks contribute equally to the gradient.

## 4. Experiments

### 4.1. Dataset Construction.

Due to confidentiality around cutting-edge process nodes, we validate on the open-source ASAP7 7nm FinFET PDK (Clark et al., 2016). A fully automatic SMT (satisfiability modulo theories)-based standard cell generator (Guo & Lin, 2025) produces >19.5k cells across 149 cell types (considering both cell function and drive strength). The dataset details are shown in Table 1. Layout resistance and capacitance are extracted with the commercial R/C extraction tool

| Cell Function | #Cell Types | #Total Variants | Example Cells |
|---|---|---|---|
| AND | 10 | 1375 | AND2D2, AND4D2 |
| OR | 10 | 1352 | OR2D1, OR3D2 |
| NAND | 10 | 1318 | NAND4D1, NAND3D2 |
| NOR | 9 | 1192 | NOR2D1, NOR3D2 |
| XOR/XNOR | 16 | 2238 | XOR2D1, XNOR2D2 |
| AO/OA | 94 | 12070 | AO33D1, OA22D1 |

*Table 1.* Statistics of the proposed dataset: cell function families, type counts, total variant counts, and representative examples

and back-annotated into the netlist. A commercial circuit simulator then characterizes the six delay/power targets to form the golden library used for labels. Stratified splits by cell type support fair regression and ranking benchmarks.

### 4.2. Experimental Setup

**Baselines.** To validate the effectiveness of our topology-guided fusion, we compare against three distinct categories of baselines: (i) **Vision-only (DeiT)**: Uses only the layout image backbone. Image tokens are average-pooled and fed directly to a regression head, ignoring netlist topology entirely. (ii) **Late Fusion**: Encodes layout and netlist via two independent encoders (identical to FusionCell's backbones). Each branch produces a global vector via mean pooling, which are then concatenated and passed to a regression MLP. No cross-attention or message passing occurs between modalities. (iii) **Symmetrical Fusion**: A dual-stream architecture using the same encoders as FusionCell but with bidirectional cross-attention (graph tokens query image tokens, and vice versa). The attended streams are pooled and concatenated, allowing both modalities to influence the representation symmetrically without a single directed query path. (iv) **ProtoCellLayout** (Luo et al., 2025): A state-of-the-art method using layout-augmented graphs. Since the code is not public, we re-implemented it by closely following the paper's embedding design.

**Metrics.** We report Mean Absolute Percentage Error (MAPE) and Coefficient of Determination ($R^2$) for regression accuracy, and Spearman's $\rho$ / Kendall's $\tau$ for ranking correlation (computed per cell type to assess within-family ordering).

**Implementation Details.** We implement FusionCell using PyTorch and train on a single NVIDIA GPU. The vision backbone is initialized with pretrained weights from `deit-base-distilled-patch16-224`. For the graph branch, we employ a 2-layer Graph Transformer (hidden dim 384, 4 heads, FFN expansion 4×, dropout 0.1) with explicit edge-type bias embeddings (initialized to zero) to capture structural information. The fusion module uses a standard multi-head cross-attention layer (embed dim 384, 4 heads) where graph tokens serve as queries and layout tokens as keys/values. All graph and fusion layers are ini-

*Table 2.* Comparison of Regression Accuracy (MAPE $\downarrow$ / $R^2$ $\uparrow$) across six targets.

| Method | Rise Delay | | Fall Delay | | Rise Trans. | | Fall Trans. | | Rise Power | | Fall Power | | Average | |
|---|---|---|---|---|---|---|---|---|---|---|---|---|---|---|
| | MAPE | $R^2$ | MAPE | $R^2$ | MAPE | $R^2$ | MAPE | $R^2$ | MAPE | $R^2$ | MAPE | $R^2$ | MAPE | $R^2$ |
| Vision-only | 2.58% | 0.926 | 2.15% | 0.897 | 3.60% | 0.943 | 3.11% | 0.904 | 4.03% | 0.960 | 3.51% | 0.959 | 3.16% | 0.947 |
| Late Fusion | 1.78% | 0.957 | 1.54% | 0.964 | 2.73% | 0.952 | 2.42% | 0.936 | 2.91% | 0.985 | 2.61% | 0.982 | 2.33% | 0.969 |
| Symmetrical Fusion | 2.17% | 0.960 | 1.80% | 0.962 | 3.17% | 0.956 | 2.80% | 0.944 | 1.73% | 0.983 | 1.47% | 0.983 | 2.19% | 0.970 |
| ProtoCellLayout | 2.91% | 0.895 | 3.77% | 0.899 | 4.93% | 0.924 | 3.64% | 0.911 | 4.81% | 0.925 | 4.90% | 0.957 | 4.16% | 0.940 |
| FusionCell (w/o Corr.) | 1.38% | 0.968 | 1.01% | 0.971 | 2.06% | 0.958 | 1.67% | 0.949 | 1.31% | **0.987** | 1.27% | 0.985 | 1.45% | 0.975 |
| **FusionCell (Ours)** | **0.94%** | **0.969** | **0.64%** | **0.975** | **1.31%** | **0.961** | **0.97%** | **0.961** | **0.86%** | 0.986 | **0.81%** | **0.986** | **0.92%** | **0.977** |

*Table 3.* Comparison of Ranking Correlation (Spearman's $\rho$ $\uparrow$ / Kendall's $\tau$ $\uparrow$) across six targets.

| Method | Rise Delay | | Fall Delay | | Rise Trans. | | Fall Trans. | | Rise Power | | Fall Power | | Average | |
|---|---|---|---|---|---|---|---|---|---|---|---|---|---|---|
| | $\rho$ | $\tau$ | $\rho$ | $\tau$ | $\rho$ | $\tau$ | $\rho$ | $\tau$ | $\rho$ | $\tau$ | $\rho$ | $\tau$ | $\rho$ | $\tau$ |
| Vision-only | 0.50 | 0.39 | 0.56 | 0.43 | 0.42 | 0.32 | 0.42 | 0.32 | 0.73 | 0.62 | 0.75 | 0.63 | 0.56 | 0.45 |
| Late Fusion | 0.78 | 0.66 | 0.81 | 0.70 | 0.72 | 0.60 | 0.76 | 0.63 | 0.89 | 0.80 | 0.88 | 0.79 | 0.81 | 0.70 |
| Symmetrical Fusion | 0.79 | 0.68 | 0.81 | 0.70 | 0.72 | 0.60 | 0.77 | 0.64 | 0.90 | 0.81 | 0.88 | 0.78 | 0.81 | 0.70 |
| ProtoCellLayout | 0.29 | 0.22 | 0.15 | 0.11 | 0.52 | 0.41 | 0.21 | 0.16 | 0.51 | 0.40 | 0.12 | 0.10 | 0.30 | 0.23 |
| FusionCell (w/o Corr.) | 0.83 | 0.72 | 0.82 | 0.73 | 0.74 | 0.62 | 0.77 | 0.65 | 0.89 | 0.82 | 0.89 | 0.81 | 0.82 | 0.73 |
| **FusionCell (Ours)** | **0.84** | **0.74** | **0.87** | **0.77** | **0.80** | **0.67** | **0.83** | **0.71** | **0.92** | **0.83** | **0.92** | **0.84** | **0.86** | **0.76** |

tialized with default PyTorch schemes (Xavier/Kaiming), except for the zero-initialized edge bias. We train the model for 50 epochs with a batch size of 8 and a learning rate of $5 \times 10^{-5}$ (AdamW optimizer). We use a stratified 9:1 random split by cell type (validation ratio 0.1) for the main experiments to ensure balanced coverage across all functional families.

### 4.3. Main Results

**Regression Accuracy.** Table 2 summarizes the regression performance. FusionCell achieves an average MAPE of $0.92\%$ and an $R^2$ of $0.977$, demonstrating exceptional precision and goodness of fit. The failure modes of the baselines offer deeper insights into the problem structure. **ProtoCellLayout** suffers the highest error (4.16%) and lowest $R^2$ (0.940), confirming that simplifying layout geometry into graph node attributes discards critical spatial information—specifically, the complex coupling capacitances that dominate delay at 7nm. **Vision-only** (3.16%) fails because of the *semantic gap*: without netlist guidance, the encoder cannot distinguish between electrically active signal tracks and floating dummy fills, treating them as identical visual textures. **Late Fusion** (2.33%) improves slightly but remains limited by its *spatial disconnect*; compressing the entire layout into a single vector prevents the model from resolving net-specific R/C environments. **Symmetrical Fusion** (2.19%) attempts to bridge this gap but falls short due to *modality confusion*: without a directed query mechanism, the attention map becomes noisy, mixing features from unrelated nets. FusionCell's superior performance validates our contribution: by treating the netlist as the "anchor" that actively queries the layout, we enforce a physically-grounded alignment that mimics the R/C extraction process itself.

**Ranking Consistency.** Accurate ranking is paramount for cell selection in synthesis flows. As shown in Table 3, FusionCell achieves state-of-the-art ranking consistency ($\rho = \mathbf{0.86}$). The poor performance of **Vision-only** ($\rho \approx 0.56$) and **ProtoCellLayout** ($\rho \approx 0.30$) highlights a key challenge: distinct cell variants often share nearly identical visual statistics (e.g., metal density) and graph topologies. A model that relies on global statistics or coarse abstractions inevitably fails to distinguish these fine-grained structural differences. FusionCell, conversely, maintains high correlation even for subtle variations. This confirms that our **heterogeneous graph modeling** and **topology-guided fusion** successfully capture the causal link between geometric changes (e.g., wire length, spacing) and performance shifts. This capability effectively positions FusionCell as a reliable surrogate for circuit simulation in ranking-sensitive tasks like drive-strength selection and layout optimization. High ranking correlation is particularly valuable for identifying Pareto-optimal cells. Instead of characterizing the entire library, designers can screen candidates with FusionCell and only run expensive sign-off on the predicted Pareto frontier, drastically reducing computational cost.

**Runtime Efficiency.** FusionCell completes inference for the entire 19.5k cell dataset in approximately 67 seconds on a single GPU. In contrast, golden label generation via traditional simulation required over 5 days. This dramatic acceleration enables rapid design space exploration that was previously infeasible.

### 4.4. Ablation Studies

We conduct comprehensive ablations to validate our design choices, focusing on the fusion strategy and netlist representation.

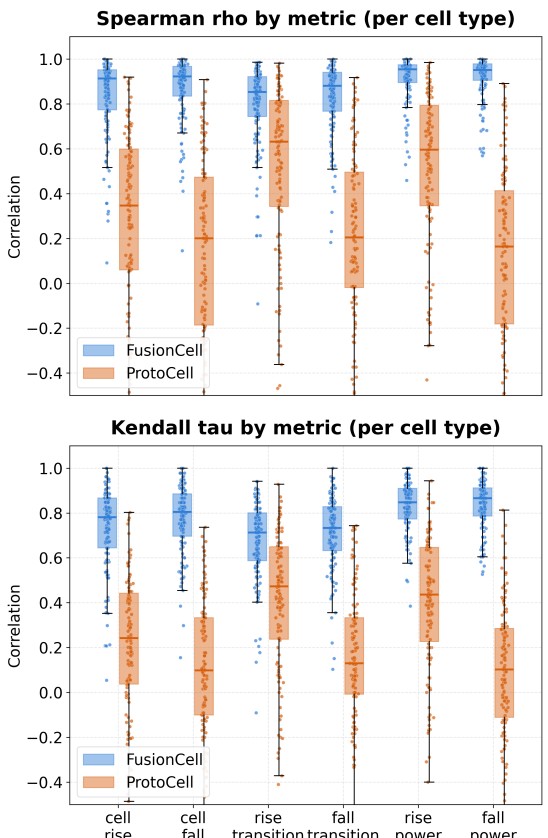

*Figure 5.* **Ranking Correlation Distribution.** Boxplots of Spearman's $\rho$ (top) and Kendall's $\tau$ (bottom) for FusionCell vs. ProtoCellLayout. FusionCell consistently maintains high correlation and low variance.

**A1: Topology-Guided vs. Symmetrical Fusion.** This ablation validates our core hypothesis: EDA fusion must be asymmetric. Tables 2 and 3 show that **Symmetrical Fusion** (MAPE 2.19%) offers negligible improvement over **Late Fusion** (MAPE 2.33%) and fails to improve ranking ($\rho = 0.81$ for both). This suggests that without a directed query mechanism, bidirectional cross-attention fails to establish meaningful correspondence, effectively degenerating into noise. In contrast, FusionCell's **Graph-Query, Layout-Key** design enforces a strong inductive bias, compelling the model to find specific physical realizations for each electrical component. This directed search aligns with physical reality—R/C effects are properties of interconnects—resulting in superior accuracy (MAPE 0.92%) and ranking ($\rho = 0.86$).

**A2: Impact of Correlation Edges.** Removing correlation edges ("FusionCell w/o Corr.") degrades MAPE (0.92% → 1.45%) and ranking $\rho$ (0.86 → 0.82). This confirms that standard connectivity edges alone are insufficient. Correlation edges effectively "short-circuit" the graph distance between structurally related nets (e.g., those sharing common functional blocks or spatial proximity), allowing the model to

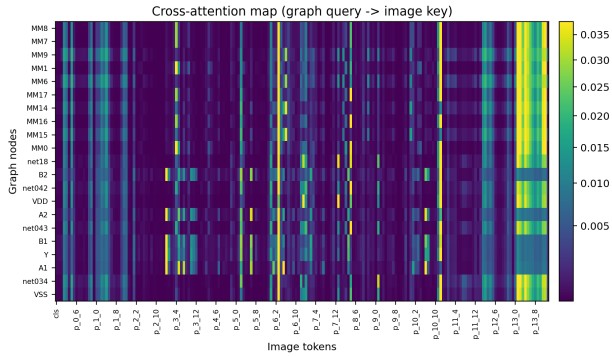

*Figure 6.* **Directed Cross-Attention Map.** Graph-query → image-key attention for the OA22x1 cell, where rows denote graph nodes and columns denote layout tokens.

capture global signal flow context and potential crosstalk interference that local message passing would miss.

**Notes on topology-only baselines.** A purely topology-only predictor is ill-posed for *within-family* layout variants that share identical netlists but differ in routed geometry. We therefore exclude it as a main baseline for variant ranking.

## 5. Discussion

**Physical Interpretability.** FusionCell offers a highly interpretable paradigm that mirrors the physical characterization flow. The **topology-guided layout attention** effectively models the **R/C extraction** process: by using net tokens to query spatial features, the model learns to identify the specific metal geometries that contribute to a net's parasitic load. Subsequently, the **fusion module**, by integrating these geometry-aware features with the topological backbone, enables the prediction of signal propagation, effectively mimicking the **circuit simulation** (ODE solving) outcome. This clear physical mapping explains why FusionCell outperforms symmetrical baselines (MAPE 0.92% vs. 2.19%). In contrast, ProtoCellLayout aggregates layout cues into graph edges via heuristic projection, lacking this explicit query-and-propagate mechanism, which leads to inferior handling of complex coupling effects.

Figure 6 further visualizes the directed Graph-Query → Image-Key cross-attention for the OA22x1 cell. The rows correspond to heterogeneous graph nodes, including device nodes such as `MM7` and `A2` and net nodes such as `net034` and `net042`; the columns correspond to spatial image tokens generated by the layout encoder, and the color intensity represents the attention weight. The map demonstrates node-specific spatial localization: the input device node `A2` concentrates on layout token `p_6_3` (weight ∼0.34), whereas internal net nodes such as `net042` and `net034` focus primarily on `p_10_12` (weights ∼0.19 and ∼0.15, respectively). It also reveals shared attention in dense metal

regions, where tokens in the `p_13` region, including `p_13_1`, `p_13_2`, and `p_13_4`, receive distributed attention from nodes including `MM7`, `net042`, `MM14`, and `net034`. Physically, this region corresponds to dense metal routing where tightly packed wires induce complex parasitic coupling, so the same geometric area can influence the R/C characteristics and performance of several surrounding nets. This behavior aligns with circuit physics and supports the intended physical interpretability of FusionCell's topology-guided multimodal fusion.

**Generalization and Application.** Beyond accuracy, this physically-grounded architecture yields robust ranking ($\rho >$ 0.86), enabling designers to rapidly screen Pareto-optimal cells before time-consuming simulation. It is worth noting that FusionCell focuses on **within-family generalization**—predicting the performance of new layout variants for known cell types. This scope aligns with the industrial reality where the set of standard cell functions (e.g., NAND, NOR, DFF) is universal and fixed across process nodes, making layout-variant generalization the primary bottleneck for characterization acceleration.

## 6. Conclusion

We presented FusionCell, a dual-modality framework for standard-cell performance prediction that employs a topology-guided fusion strategy. By leveraging the netlist to actively query layout geometry, FusionCell enforces a physically-grounded alignment between electrical topology and physical realization. Experiments on a 7nm benchmark show that FusionCell achieves superior regression accuracy (MAPE 0.92%) and ranking consistency ($\rho \approx 0.86$) compared to vision-only and symmetrical baselines. This work establishes a generalizable paradigm for multimodal EDA, aligning model architecture with the physical hierarchy of design data. Future work will extend this approach to more advanced process nodes with complex design rules and multi-corner characterization. The open-source repository will be available at https://github.com/zhywhite/PreCell.

## Impact Statement

This work introduces FusionCell, which accelerates standard-cell performance prediction by $10^4\times$ compared to traditional simulation. By replacing days of heavy computation with millisecond-level inference, FusionCell significantly reduces the energy consumption and carbon footprint of the chip design process. Furthermore, it enables rapid design-space exploration, empowering researchers to develop more energy-efficient hardware and fostering the democratization of advanced EDA tools.

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
