# OpenReview forum: "FusionCell: Cross-Attentive Fusion of Layout Geometry and Netlist Topology for Standard-Cell Performance Prediction"
_ICML.cc/2026/Conference — ICML 2026 regular_

### Official Review · Reviewer_WQTj · 2026-03-09

**Soundness:** 3
**Presentation:** 3
**Significance:** 2
**Originality:** 2
**Overall Recommendation:** 4
**Confidence:** 5

**Summary:**

This paper proposes a learning based method, called, FusionCell, for efficient characterization of standard cell libraries. The main claim made by the authors, which has been experimentally demonstrated to be efficient is that FusionCell leverages cross-attentive fusion of the features from  layout geometry and cell netlist, something that has not been explored in prior methods.

**Compliance With Llm Reviewing Policy:**

Affirmed.

**Key Questions For Authors:**

* While the authors show that the presented topology-guided graph-query/image-key cross-attention outperforms more expensive symmetric cross-attention, the authors should provide more in-depth insights on why this is happening. Does layout-guided cross-attention alone work?

* The authors do not provide information how what training data is used to train the model and training time.

* Claiming that "FusionCell completes inference for the entire 19.5k cell dataset in approximately 67 seconds on a single GPU. In contrast, golden label generation via traditional simulation required over 5 days" is somewhat misleading without providing the cost of training.

**Strengths And Weaknesses:**

**Strengths**:

* This paper is well written, very easy to understand and follow; it clearly shows the main ideas and important implementation details.
* The work presented constitutes a good ML application, which targets standard cell characterization, a key task in digital IC design.
* It makes sense to jointly consider both the layout and netlist features and fuse them properly as done in the proposed FusionCell approach.
* One interesting design and observation of the FusionCell architecture is the presented topology-guided graph-query/
image-key cross-attention, which has been shown to be more effective than symmetric cross-attention,.

**Weaknesses**:
 * While being a good ML application papers, most aspects of this work are pretty standard adoption of well-established existing models, DeiT for layout, and a graph transformer for netlist.

* This work lacks fundamental new ML contributions and deep domain-specific adaption of existing ML techniques.

* The topology graphs incorporate additional "correlation" edges - however it is not clear on what principles these edges are introduced. Also, are the introduced manually?

---

> ### Author Rebuttal · Authors · 2026-03-27
>
> We thank the reviewer for the positive assessment and detailed feedback. We are encouraged that the reviewer recognizes the clarity, practical relevance, and effectiveness of the proposed topology-guided fusion.
>
> R.Q1: We thank the reviewer for this insightful question. We find that layout-guided cross-attention (using layout as query) also performs worse in this setting, with an average 1.3% increase on MAPE. The key reason is that layout tokens alone do not carry explicit information about electrical roles, which are the fundamental determinants of final circuit performance. In standard cell design, the mapping from layout to performance is highly degenerate—distinct layout geometries can often yield very similar performance metrics. Conversely, different topologies exhibit clear, discriminative performance boundaries. Therefore, relying solely on layout features cannot effectively guide feature aggregation. Instead, topological features, which encapsulate critical electrical roles, serve as a far superior and physically meaningful query to extract relevant layout contexts. Meanwhile, topology-guided attention aligns with the physical characterization process: the netlist provides the structural “skeleton”, while layout determines the parasitic perturbations. Using topology as the query allows the model to retrieve geometry conditioned on specific electrical functions, which better reflects how delay/power are determined in practice. Therefore, topology-guided querying enables more structured and physically meaningful feature aggregation, leading to improved performance over both symmetric and layout-guided alternatives. We will clarify this intuition more explicitly in the revision.
>
> R.Q2: We agree that this should be stated more clearly. The model is trained on the ASAP7-based dataset described in the paper (roughly 19.5k standard cells across 149 types). On a single RTX 4090, training for 50 epochs takes less than 10 hours. We will add these details explicitly in the revision.
>
> R.Q3:  We agree that inference speed should be interpreted together with training cost. As noted above, training the model takes less than 10 hours on a single RTX 4090. Importantly, this is a one-time cost, after which the trained model can be reused to evaluate large numbers of cell variants. Therefore, the reported speedup reflects the practical deployment scenario: once trained, FusionCell enables rapid evaluation (e.g., ~67 seconds for the full dataset), compared to circuit simulation that can take hours to days for characterization. We will clarify this distinction more explicitly in the revision.
>
> R.W3: Regarding correlation edges, these are automatically constructed based on structural relationships between nets, intended to model parasitic interactions (e.g., coupling effects) that are physically grounded and not captured by standard device–net edges. This design further strengthens the model's alignment with real circuit behavior and enhances its predictive capability.
>
> Summary: We thank the reviewer again for the constructive feedback. We will clarify the design motivation of topology-guided fusion, provide training cost details, and improve the discussion on efficiency and correlation edges in the revision. We emphasize that the core contribution lies in physically grounded multimodal modeling, rather than generic architectural novelty, which is critical in this domain. We hope these clarifications further strengthen the paper and would greatly appreciate the reviewer’s reconsideration of the score.

---

> > ### Author Rebuttal · Reviewer_WQTj · 2026-04-04
> >
> > * More details about the construction of correlation edges are needed. How is the process automated? On what principle? What is the complexity and quality of the process?

---

> > > ### Author Response · Authors · 2026-04-04
> > >
> > > We will clarify that the construction of correlation edges is a straightforward, deterministic process without complex algorithms.
> > > Automation & Principle: It is entirely rule-based and directly reflects physical layout parasitics. Because nets connected to the same cell are spatially close and exhibit mutual parasitic coupling, we simply add a correlation edge between every pair of nets within the same cell.
> > > Complexity: The overall time complexity is linear, $O(N)$, where $N$ is the total number of cells. For a cell with $k$ nets, creating edges takes $O(k^2)$ operations, but $k$ is bounded by a very small constant in standard cell libraries.
> > > Quality: The process guarantees high physical fidelity. This simple mapping explicitly injects crucial spatial and parasitic priors into our graph, directly capturing real-world layout interactions without requiring implicit learning.

---

### Official Review · Reviewer_TvsU · 2026-03-11

**Soundness:** 3
**Presentation:** 3
**Significance:** 3
**Originality:** 3
**Overall Recommendation:** 3
**Confidence:** 3

**Summary:**

The FusionCell proposed in this article is based on the Layout Geometry after routing modeling and netlist topology, and designs a dual-modal predictor for performance prediction. Additionally, a 7nm dataset was constructed using the ASAP7 PDK. Experiments show that the FusionCell proposed in this article can effectively reduce regression error while improving Spearman/Kendall ranking.

**Compliance With Llm Reviewing Policy:**

Affirmed.

**Key Questions For Authors:**

1. Does experiments on public datasets are possible? Can you provide the experimental results?
2. Are you considering open-sourcing simulation datasets？

**Limitations:**

yes

**Strengths And Weaknesses:**

Strengths: 1. The proposed FusionCell shows good exceptional prediction accuracy and ranking consistency on tested benchmarks; 2. The proposed FusionCell efficiently computes the predicted results, which shows industrial potential; 3. The proposed dual-modality modeling method exhibits a certain degree of innovation. Weaknesses: 1. The experiment was conducted on an unpublished dataset, which makes it somewhat lacking in persuasiveness; 2. Patch size in DeiT directly affects the model's accuracy in capturing thin metal lines and outlier samples ( like units with abnormally large area or extremely congested wiring ) should be considered. Thus, related ablation studies should be conducted.

---

> ### Author Rebuttal · Authors · 2026-03-26
>
> We thank the reviewer for the positive evaluation of our work, especially regarding the strong predictive accuracy, ranking consistency, and industrial relevance. We also appreciate the constructive suggestions. The main responses are as follows:
>
> R.Q1:
> We fully agree that evaluation on public datasets would strengthen the work. However, to the best our knowledge, there is currently no publicly available dataset that jointly provides routed layout geometry, netlist topology, and golden delay/power labels for standard-cell characterization at this scale. Even closely related work such as ProtoCellLayout does not release any code or dataset, which reflects the general difficulty of building such datasets in this domain. Our dataset is built on the open ASAP7 PDK with a fully automated pipeline, and is therefore reproducible in principle. We acknowledge that releasing a benchmark would be highly valuable to the community. We plan to release a sanitized version of the dataset and generation pipeline upon acceptance and actively promote open benchmarking in this direction. We hope this effort can help establish a common testbed for future research.
>
> R.Q2 :
> We appreciate this important question. We will release the dataset and code in the final manuscript. While ASAP7 is an open-access PDK, certain high-fidelity simulation parameters and setups cannot be freely redistributed. Our plan is to release desensitized layout representations, netlist graph data, and the data generation pipeline. For the golden simulation labels, we will provide the necessary setup and instructions, allowing users to regenerate the labels with access to the appropriate environment. We believe this approach strikes a balance between openness and the constraints of advanced-process characterization.
>
> R.W2. (weakness):
> We agree that patch size is an important factor for capturing fine-grained routing patterns, especially thin metal lines and outlier cases such as unusually large-area or highly congested cells. We have conducted additional analysis (not included due to rebuttal constraints): compared with the current 16×16 setting, both 8×8 and 32×32 patches lead to roughly 0.5% worse performance on our metrics. Intuitively: 8×8 preserves more local detail, but introduces higher noise and computational overhead 32×32 is too coarse and loses important fine-grained geometric information
> 16×16 provides the best trade-off for typical standard-cell library layouts: it is neither overly fine nor overly coarse
> We will include a dedicated patch-size ablation and robustness discussion in the revision.
>
> We thank the reviewer again for the helpful feedback. In the revision, we will: clarify the availability and limitations of public data for this task make our best effort toward releasing shareable resources and promoting open benchmarking include patch-size ablation and robustness analysis We hope these clarifications address the reviewer’s concerns, and we would greatly appreciate the reviewer’s reconsideration of the score.

---

> > ### Author Rebuttal · Reviewer_TvsU · 2026-04-04
> >
> > Thanks to the authors for the detailed rebuttal.
> >
> > I appreciate you running the additional ablation study on the DeiT patch sizes (8x8, 16x16, 32x32).
> >
> > Regarding the dataset, I understand the inherent challenges in this domain and the licensing constraints associated with the ASAP7 PDK. While I appreciate your future commitment to releasing a sanitized dataset and the generation pipeline, the fact remains that the data and the golden simulation labels cannot be fully verified or benchmarked during this current review phase.
> >
> > Overall, I still think this is a solid piece of work with industrial potential. However, because my primary concern about dataset verifiability and persuasiveness at this stage remains, I will be keeping my original score.

---

> > > ### Author Response · Authors · 2026-04-04
> > >
> > > We sincerely thank the reviewer for the recognition of our work. After a thorough legal review, we confirm that we will release the complete dataset, golden labels, and full source code in the final version.
> > >
> > > To address the verifiability concern during this review phase, we have provided an anonymous repository with representative data samples:
> > >
> > > Anonymous Repository: https://anonymous.4open.science/status/dataset-4D9B
> > > (Note: Due to the massive size of the full dataset, we have uploaded essential samples for verification first; the complete repository will be hosted on a dedicated platform upon publication.)
> > >
> > > As the primary concern regarding dataset transparency and verifiability has been addressed, we kindly hope the reviewer will reconsider the score. We believe these resources, combined with our work's industrial potential, resolve the remaining concerns. We greatly appreciate your time and rigor.

---

### Official Review · Reviewer_7YXw · 2026-03-12

**Soundness:** 3
**Presentation:** 4
**Significance:** 3
**Originality:** 3
**Overall Recommendation:** 4
**Confidence:** 3

**Summary:**

This paper proposes FusionCell, a dual-modality framework for standard-cell performance prediction, targeting delay, transition, and power estimation from both routed layout geometry and netlist topology. The central design choice is a topology-guided multimodal fusion mechanism: instead of symmetric cross-attention or simple late fusion, the model uses heterogeneous graph tokens derived from the netlist as queries to attend to spatial layout tokens extracted from a DeiT-based image encoder over multi-layer routed layouts. The paper argues that this asymmetric design better matches the physical causality of standard-cell behavior, where functional connectivity provides an anchor for interpreting layout-dependent parasitics. Experiments on a large ASAP7-based 7nm dataset with roughly 19.5k cells show very strong predictive accuracy, including sub-1% average MAPE and high ranking correlation, while also offering major runtime advantages over circuit simulation.

**Compliance With Llm Reviewing Policy:**

Affirmed.

**Key Questions For Authors:**

1. The graph module appears to rely on global node-to-node attention. Can the authors comment on the computational and memory complexity as graph size increases? While this is likely manageable for standard cells, what happens if the method is extended to larger circuit fragments?

2. The runtime comparison is primarily made against traditional circuit simulation. Could the authors provide a more direct comparison of inference latency, throughput, and memory usage against the machine-learning baselines included in the experiments?

3. The paper argues that the fusion mechanism is physically interpretable. Could the authors provide qualitative visualizations of cross-attention maps or token-to-region correspondence to support this claim more directly?

4. How sensitive is the model to the exact rasterization strategy for representing routed layout layers? For example, would modest changes in resolution, layer encoding, or via projection materially affect performance?

5. Is the model trained and evaluated only at a fixed PVT corner? If so, how do the authors see the approach extending to multi-corner or multi-condition prediction?

6. Can the authors discuss the model’s behavior under distribution shift, such as unseen cell families, routing heuristics, or modified library design styles?

7. Since ranking quality is especially important for screening tasks, could the authors provide more analysis on failure modes in ranking, for instance where two cells have close ground-truth performance but are misordered by the model?

**Limitations:**

yes

**Strengths And Weaknesses:**

Strengths

1. Strong problem relevance and practical significance.
Accurate and fast standard-cell performance prediction is an important problem for modern design technology co-optimization, cell library exploration, and circuit implementation workflows. The ability to replace or reduce expensive SPICE characterization with a learned surrogate has clear practical value, especially when many candidate layout variants must be screened quickly.

2. The multimodal formulation is physically well motivated.
A major strength of the paper is that the fusion mechanism reflects domain structure rather than treating the two modalities symmetrically by default. Using topology-derived graph tokens to query layout features is a sensible inductive bias: the netlist defines the logical interactions, while the routed layout determines the physical realization and parasitic effects. This is more convincing than a generic “fuse everything and let attention figure it out” approach.

3. Clear presentation.
The paper is generally well written and easy to follow. The architectural overview is understandable, and the explanation of the multi-layer routed-layout representation is intuitive. Compared with many AI4EDA submissions, this paper does a good job of making the pipeline legible.

Weaknesses

1. Insufficient computational cost comparison against ML baselines.
The paper emphasizes speedup versus circuit simulation, which is useful, but that is only part of the deployment story. It would also be helpful to understand the training/inference overhead, memory footprint, and model complexity relative to other learned baselines such as layout-only or prior multimodal methods.

2. Generalization scope is somewhat narrow.
The experimental setting appears focused on one technology node and one dataset family. The results are strong within that regime, but the paper would be stronger if it offered more evidence about transfer across cell families, routing styles, technology assumptions, or corner conditions.

3. Limited treatment of robustness across PVT or corner variation.
Since standard-cell characteristics are highly dependent on process, voltage, and temperature conditions, it would be useful to clarify whether the model is intended only for a fixed corner, or whether the approach can be extended to multi-corner prediction.

---

> ### Author Rebuttal · Authors · 2026-03-27
>
> We thank the reviewer for the positive evaluation and for recognizing the practical relevance and physically motivated design of our approach.
>
> R.Q1: Our current Graph Transformer uses full attention, with per-layer complexity scaling as 𝑂(𝐻⋅𝑁^2), where 𝑁 is the number of nodes (devices + nets). In our setting, standard-cell graphs are small (typically tens of nodes), so N^2 remains on the order of a few thousand, and both computation and memory are well within practical GPU limits. We agree that directly extending this design to larger circuit fragments would lead to quadratic scaling in both compute and memory (e.g., attention logits and bias tensors), which can become a bottleneck. However, standard-cell circuits typically contain fewer than 100 transistors, so this scaling issue is not a concern for standard-cell characterization. To scale beyond standard cells, methods such as restricting the encoder to local subgraphs or adopting sparse/block-sparse attention would be required.
>
> R.Q2: We have measured direct efficiency against ML baselines and will include these results in the revision. In our current implementation, FusionCell runs at batch size 32 with 0.00364 s/sample (274.9 samples/s), 0.115 s/batch, and 635 MB peak GPU memory. In comparison, our reproduced ProtoCellLayout runs at batch size 512 with 0.00090 s/sample (1107.0 samples/s), 0.433 s/batch, and 2150.7 MB peak GPU memory. Compared with circuit simulation that typically takes hours or days per library characterization, this second-level ML overhead is negligible in practice, considering the prediction quality.
>
> R.Q3: Due to rebuttal constraints, we cannot include additional figures here. As also noted in our response to Reviewer 4 (Q3), we will include cross-attention visualizations in the revision.
>
> R.Q4: We agree that the rasterization strategy is an important design choice. For resolution, we find the model robust to moderate changes, while overly fine or overly coarse settings both degrade performance by about 0.5%. Intuitively, higher resolution preserves more local routing detail but increases noise and computational overhead, whereas lower resolution is too coarse and loses fine-grained geometric information. For layer encoding, we observe little impact, since the key structural information is already carried by the geometric patterns across the routed layers rather than the exact encoding choice itself. For via projection, the effect is also relatively limited, as it mainly changes how cross-layer connectivity is exposed in the rasterized view rather than the underlying layout/netlist structure. We will include a dedicated ablation and robustness discussion on rasterization choices in the revision.
>
> R.Q5: We confirm that the current model is trained and evaluated at a fixed PVT corner. We note that for a given layout, the underlying R/C parasitic characteristics are determined by geometry and remain largely unchanged across PVT conditions. The primary effect of PVT variation is on the final simulation outcomes (e.g., delay/power), which can be viewed as a systematic shift rather than a change in structural features. Therefore, we expect the approach to extend naturally to multi-corner settings. In practice, this can be achieved by: fine-tuning the model with a relatively small amount of data from new PVT conditions, or
> conditioning the model on PVT parameters for multi-task prediction in future work.
>
> R.Q6:  We agree that robustness under distribution shift is an important consideration. In practice, standard-cell libraries share highly similar netlist structures and design conventions, so truly “unseen” cell families are relatively limited. Most variations arise from layout styles or routing heuristics rather than fundamentally new topologies. Our approach is designed to align with the underlying physical process: it models geometric layout features and netlist connectivity, which are the primary factors determining parasitic behavior. As a result, we expect good transferability across moderate variations in routing style or library design. For more significant shifts (e.g., new design styles or corner cases), the model can be efficiently adapted via lightweight fine-tuning with a small amount of new data, since the core geometric and topological representations remain unchanged.
>
> R.Q7: Ranking metrics such as Spearman’s ρ and Kendall’s τ are strict order-based measures. In practice, we observe that most misordered cases occur when two cells have very close ground-truth performance. In such cases, even a small prediction error (e.g., within ~1%) can lead to a reversal in ordering, despite both predictions being numerically accurate. Therefore, these ranking inconsistencies are often due to intrinsically small performance gaps, rather than large prediction errors.
>
> We thank the reviewer again for the supportive feedback, and hope our clarifications further strengthen the case for acceptance.

---

### Official Review · Reviewer_9ucL · 2026-03-12

**Soundness:** 3
**Presentation:** 3
**Significance:** 3
**Originality:** 2
**Overall Recommendation:** 4
**Confidence:** 3

**Summary:**

FusionCell predicts standard-cell delay and power by fusing routed layout geometry (via DeiT on 3-channel metal-layer images) with netlist topology (via graph transformer on heterogeneous device–net graphs). The core mechanism is topology-guided cross-attention, where netlist tokens query layout tokens. On a 7nm ASAP7 dataset of 19.5k cells (149 types), FusionCell achieves 0.92% average MAPE and 0.86 Spearman ρ across six delay/power targets.

**Compliance With Llm Reviewing Policy:**

Affirmed.

**Final Justification:**

The rebuttal solved all my main concerns, so I will change my score from 3 to 4.

**Key Questions For Authors:**

1) Can you provide a topology-only baseline to isolate the contribution of layout information?

2) Can you validate the ProtoCellLayout re-implementation by reproducing their reported numbers on their original dataset?

3) Can you show attention maps demonstrating that netlist tokens attend to physically meaningful layout regions?

4) Have you analyzed correlation among the six targets? If rise/fall metrics are highly correlated, the effective dimensionality of the prediction task is lower than suggested.

5) How does FusionCell perform on cells from a different layout generation tool or PDK?

**Limitations:**

yes

**Strengths And Weaknesses:**

Strengths

1) The asymmetric fusion design—netlist as query, layout as key/value—has clear physical motivation: the netlist defines what electrical structures to look for, while the layout provides the spatial R/C evidence. This mirrors the actual R/C extraction process.

2) The 19.5k-cell dataset on ASAP7 PDK with automated generation is a useful resource. The six-target formulation (rise/fall delay, transition, power) is comprehensive for standard-cell characterization.

Weaknesses

1) Only one true external baseline (ProtoCellLayout), and it is re-implemented since code is unavailable. Its anomalously poor ranking (ρ ≈ 0.30) raises serious fairness concerns. The other three "baselines" (vision-only, late fusion, symmetrical fusion) are ablation variants of FusionCell itself. By ICML standards, this comparison is insufficient.

2) No topology-only baseline is provided—the paper itself acknowledges this omission. Without it, the marginal contribution of layout information cannot be quantified. Additionally, no attention visualization is shown to support the central claim that netlist tokens attend to electrically relevant layout regions.

3) All 19.5k cells are generated by a single SMT-based tool on a single PDK. Layout diversity is likely limited. Cross-PDK or cross-tool generalization is completely untested.

4) Topology-guided cross-attention is standard multimodal fusion (Q from one modality, K/V from another). The domain-specific insight is valuable but the technical contribution is incremental.

---

> ### Author Rebuttal · Authors · 2026-03-27
>
> We thank the reviewer for the careful reading and constructive feedback. We are encouraged that the reviewer recognizes the physical motivation of our asymmetric fusion design and the value of the dataset. We respond to the reviewer’s comments as follows.
>
> R. Q1: We appreciate the reviewer’s suggestion. Actually, under the task setting studied in this paper, a topology-only baseline is not a well-posed learning problem. The reason is that, in our dataset, multiple routed layout variants share the same netlist but can still differ by up to 10% in delay/power, due to layout-dependent parasitic effects introduced by routing geometry. Therefore, when only netlist information is provided, the mapping from input to target is one-to-many rather than one-to-one. In other words, the target performance is not uniquely determined by topology alone. As a result, a topology-only predictor cannot, in principle, resolve within-family layout-dependent variation and would collapse toward predicting an average over variants with the same netlist. For this reason, we did not include it as a main baseline, since it would not constitute a meaningful or fair predictor for the problem formulated in this paper.
> We will clarify this point more explicitly in the revision.
>
> R. Q2.:  We would like to clarify that the original ProtoCellLayout work does not release its dataset or implementation, which makes exact reproduction on their original setup infeasible. Nevertheless, we made our best effort to follow the method as described in the paper, including the graph construction and layout encoding strategy.
> Importantly, we observe that: On regression metrics, our reproduced results are closely aligned with those reported in the ProtoCellLayout paper, which gives us confidence that the implementation is faithful. Specifically, their reported errors for internal power, timing delay, and transition time are 4.6%, 3.6%, and 3.9%, respectively, while our reproduced results are 4.85%, 3.34%, and 4.28%. These numbers are of the same scale and do not differ materially, suggesting that our re-implementation is broadly consistent with the original method. Regarding ranking performance, the original ProtoCellLayout paper does not report ranking metrics (e.g., Spearman/Kendall). We introduce ranking correlation as an additional evaluation criterion, which is more relevant for tasks like cell selection and Pareto screening. While direct comparison is not possible, we believe including these metrics provides a more stringent and meaningful evaluation of model performance.
>
> R.Q3: We agree that attention visualization is important for supporting the interpretability claim. Due to rebuttal constraints, we cannot include additional figures here, but we will add cross-attention maps in the revision. Specifically, we will visualize how netlist tokens attend to layout regions and show representative cases where attention focuses on routing areas corresponding to queried nets, which is consistent with the intended topology-guided fusion mechanism.
>
> R.Q4: We agree that the six targets are not fully independent, since they are all derived from the same underlying electrical behavior and characterization flow. However, their relationships are complex and do not admit a simple or standard reduction in practice. Such interdependent and complex metrics are very common in chip design and manufacturing, where physically related quantities still need to be modeled separately. Moreover, in this task, even around 1% error is practically important, so we evaluate the six metrics independently rather than assuming a lower-dimensional surrogate.
>
> R.Q5: We expect limited sensitivity to the layout generation tool, as long as the layouts follow the same design rules, since the input representations (layout geometry and netlist topology) remain consistent. For PDK changes, the primary impact is on simulation outcomes (i.e., parasitic scaling), which can be viewed as a systematic shift, while the underlying layout/netlist structures are largely preserved, suggesting the model can still generalize.
> That said, we acknowledge this is an important direction. Due to limited access to additional PDKs and the high cost of regenerating full datasets, we have not yet conducted cross-PDK validation. We will clarify this limitation and leave it as future work.
>
> Summary.
> We thank the reviewer for the helpful feedback. For this task, we have already compared the methods that can be meaningfully and fairly evaluated: ProtoCellLayout and representative alternatives like vision-only, late fusion, and symmetrical fusion. A topology-only baseline is not meaningful here because one netlist can correspond to multiple routed layouts with different performance labels. Therefore, we believe the current comparisons are sufficient to support the core conclusion that physically guided multimodal fusion is necessary, and we would greatly appreciate the reviewer’s reconsideration of the score.

---

> > ### Author Rebuttal · Reviewer_9ucL · 2026-04-04
> >
> > Thanks for your great answers to my questions and concerns. Although it is allowed to show figures in the rebuttal comment, it would be helpful to see the attention figure and its associated explanation. Could you try to put your attention map in an anonymous GitHub repository?

---

> > > ### Author Response · Authors · 2026-04-04
> > >
> > > Thank you for your positive feedback and for the excellent suggestion to provide a visual breakdown of the attention mechanism. We agree that visualizing the cross-attention map significantly strengthens the interpretability of our method.
> > > We sincerely apologize for not including the figure in our initial response. We were previously under the impression that providing external links or images was not permitted during the rebuttal phase due to our unfamiliarity with the specific formatting rules. We truly appreciate you clarifying this for us!
> > > Per your request, we have now uploaded the high-resolution attention map for the OA22x1 standard cell (along with the raw attention weight logs) to an anonymous GitHub repository for your review: https://anonymous.4open.science/r/icmlrebuttal-6EE3.  We hope these clarifications address the reviewer’s concerns, and we would greatly appreciate the reviewer’s reconsideration of the score.

---

### Decision · Program_Chairs · 2026-04-30

**Decision:**

Accept (regular)

**Comment:**

In this application paper, the authors propose a method for standard cell (presumably the building block of digital circuits) performance prediction. The approach uses a tailored (graph) transformer-based architecture to predict standard cell performance from routed layout geometry and netlist topology. The proposal shows clear gains across metrics and settings.

The reviewers have indicated that this work presents an interesting ML application paper that targets an important task in digital IC design. However, concerns about the lack of machine learning-related methodological insights and limited baselines have been mentioned. The authors provide a thorough rebuttal, and several concerns have been addressed during the rebuttal or discussion phase. As this work is far outside my expertise, I follow the reviewers' recommendation and recommend weak acceptance.